# Genomic landscape and chronological reconstruction of driver events in multiple myeloma

Francesco Maura[1,2,3,14], Niccoló Bolli[3,4,14], Nicos Angelopoulos[2,5], Kevin J. Dawson[2], Daniel Leongamornlert[2], Inigo Martincorena[2], Thomas J. Mitchell [2], Anthony Fullam[2], Santiago Gonzalez [6], Raphael Szalat[7], Federico Abascal [2], Bernardo Rodriguez-Martin[8], Mehmet Kemal Samur[7], Dominik Glodzik [2,9], Marco Roncador[2], Mariateresa Fulciniti[7], Yu Tzu Tai[7], Stephane Minvielle[10], Florence Magrangeas[10], Philippe Moreau[10], Paolo Corradini[3,4], Kenneth C. Anderson[7], Jose M.C. Tubio[2,8], David C. Wedge[11], Moritz Gerstung [6], Hervé Avet-Loiseau[12], Nikhil Munshi[7,13,15] & Peter J. Campbell[2,15]

The multiple myeloma (MM) genome is heterogeneous and evolves through preclinical and post-diagnosis phases. Here we report a catalog and hierarchy of driver lesions using sequences from 67 MM genomes serially collected from 30 patients together with public exome datasets. Bayesian clustering defines at least 7 genomic subgroups with distinct sets of co-operating events. Focusing on whole genome sequencing data, complex structural events emerge as major drivers, including chromothripsis and a novel replication-based mechanism of templated insertions, which typically occur early. Hyperdiploidy also occurs early, with individual trisomies often acquired in different chronological windows during evolution, and with a preferred order of acquisition. Conversely, positively selected point mutations, whole genome duplication and chromoplexy events occur in later disease phases. Thus, initiating driver events, drawn from a limited repertoire of structural and numerical chromosomal changes, shape preferred trajectories of evolution that are biologically relevant but heterogeneous across patients.

[1] Myeloma Service, Department of Medicine, Memorial Sloan Kettering Cancer Center, New York, NY, USA. [2] The Cancer, Ageing and Somatic Mutation Programme, Wellcome Sanger Institute, Hinxton, Cambridgeshire CB10 1SA, UK. [3] Department of Medical Oncology and Hemato-Oncology, University of Milan, Milan, Italy. [4] Department of Medical Oncology and Hematology, Fondazione IRCCS Istituto Nazionale dei Tumori, Milan, Italy. [5] School of Computer Science and Electronic Engineering, University of Essex, Colchester, UK. [6] European Bioinformatics Institute, European Molecular Biology Laboratory (EMBL-EBI), Hinxton, UK. [7] Jerome Lipper Multiple Myeloma Center, Dana–Farber Cancer Institute, Harvard Medical School, Boston, MA, USA. [8] CIMUS - Molecular Medicine and Chronic Diseases Research Centre, University of Santiago de Compostela, Santiago de Compostela, Spain. [9] Epidemiology and Biostatistics, Memorial Sloan Kettering Cancer Center, New York, NY, USA. [10] CRCINA, INSERM, CNRS, Université d'Angers, Université de Nantes, Nantes, France. [11] University of Oxford, Big Data Institute, Oxford, UK. [12] IUC-Oncopole, and CRCT INSERM U1037, 31100 Toulouse, France. [13] Veterans Administration Boston Healthcare System, West Roxbury, MA, USA. [14] These authors contributed equally: Francesco Maura, Niccoló Bolli. [15] These authors jointly supervised this work: Nikhil Munshi and Peter J. Campbell. Correspondence and requests for materials should be addressed to N.M. (email: nikhil_munshi@dfci.harvard.edu) or to P.J.C. (email: pc8@sanger.ac.uk)

The genome of multiple myeloma (MM) is complex and heterogeneous, with a high frequency of structural variants (SVs) and copy-number abnormalities (CNAs)[1–3]. Translocations between the immunoglobulin heavy chain (IGH) locus and recurrent oncogenes are found in ~40% of patients. Cases without IGH translocations often have a distinctive pattern of hyperdiploidy affecting odd-numbered chromosomes, where the underlying target genes remain mysterious. These SVs and recurrent CNAs are considered early drivers, being detectable also in premalignant stages of the disease[1–3]. Cancer genes are also frequently altered by driver point mutations, with mitogen-activated protein kinase (MAPK) and nuclear factor kappa-light-chain-enhancer of activated B cells (NF-KB) signaling as major targets[4–8].

Many blood cancers develop along preferred evolutionary trajectories. Early driver events, drawn from a restricted set of possible events, differ in which subsequent cancer genes confer clonal advantage, leading to considerable substructures of co-operativity and mutual exclusivity among cancer genes. These subtypes vary in chemosensitivity and survival, suggesting that although patients share a common histological and clinical phenotype, the underlying biology is distinctly heterogeneous. Preliminary studies have suggested that these patterns exist in MM as well[5–12], but have not yet been systematically defined in large cohorts with broad sequencing coverage. There have been recent reports of dependencies among MM driver mutations using either targeted or exome-based approaches[9,13], but we lack a comprehensive characterization of MM genomic subgroups based on the complete catalog of driver mutations, copy-number changes and recurrent SVs. In addition, since the first MM whole genome sequencing study[6], the landscape of nonrecurrent SVs and complex events has not been systematically explored.

In this study, we combine a large cohort of serial MM samples analyzed by whole-genome sequencing (WGS) with a publicly available dataset to define driver events and how they group across patients, with implications for disease classification. Furthermore, we describe the temporal evolution of the disease in preclinical phases, highlighting the unexpected dynamism of genomic changes, often in the form of private, complex structural events.

## Results

**Landscape of driver mutations in MM.** We performed whole WGS of 67 tumor samples collected at different time points from 30 MM patients, together with matched germline controls (Supplementary Data 1 and 2, "Methods"). We also included in our analyses published whole exome data from 804 patients within the CoMMpass trial (NCT01454297)[14]. To discover MM driver genes, we analyzed the ratio of nonsynonymous to synonymous mutations, correcting for mutational spectrum and covariates of mutation density across the genome with the dNdScv algorithm[15–17]. Overall, 55 genes were significantly mutated with a false discovery rate of 1% (Fig. 1a and Supplementary Data 3). Our shortlist of genes showed a 65% overlap with a recently published study from the Myeloma Genome Project (MGP) (Supplementary Data 3)[13], with less frequently mutated genes accounting for the discordant calls. This is expected given the random sampling and differences in statistical approaches and power between studies (Supplementary Data 3). To confirm this, we restricted the multiple hypotheses testing p-value correction to the set of unique driver genes from the MGP (n = 26), thus identifying six additional shared drivers (ABCF1, ZFP36L1, TET2, ARID2, KDM6A, and EP300) (Supplementary Fig. 1 and Supplementary Data 3) and increasing the concordance of the two datasets. Overall, 87% of all MMs had at least one of

these 61 driver genes mutated, with an average of 2.04 (0.98–3.08) drivers per sample estimated calculating the global ratio of non-synonymous to synonymous mutations[17] ("Methods"). A significant fraction of these driver point mutations was present in subclones of the myeloma rather than the ancestral clone, suggesting that they often play a role in later phases of cancer development (Supplementary Fig. 2a).

Beyond well-known myeloma genes such as KRAS, NRAS, DIS3, and FAM46C[5–8], several other interesting candidate genes emerged. The linker histones HIST1H1B, HIST1H1D, HIST1H1E, and HIST1H2BK all showed a distinctive pattern of missense mutations clustered in the highly conserved globular domain (Supplementary Fig. 2b–e), as also reported in follicular lymphoma[18]. Many of the mutations were nearby, or directly affected, conserved positively charged residues critical for nucleosome binding, suggesting that they disrupt the histones' role in regulating higher order chromatin structure. FUBP1, an important regulator of MYC transcription[19], showed an excess of splice site and nonsense mutations, emerging as a potential tumor suppressor gene in MM (Supplementary Fig. 2f). MAX, a DNA-binding partner of MYC, showed an interesting pattern of start-lost mutations, nonsense and splice site mutations, together with hotspot missense mutations at residues Arg35, Arg36, and Arg60, known to abrogate DNA binding[7] (Supplementary Fig. 2g). Genes with rather more mysterious function were also significant: the zinc finger ZNF292, recently described as mutated in MM, chronic lymphocytic leukemia and diffuse large B-cell lymphoma[13,20,21], showed an excess of protein-truncating variants (Supplementary Fig. 2h); the uncharacterized TBC1D29 gene showed missense mutations clustered in the last exon[22] (Supplementary Fig. 2i).

**Dependencies of driver events.** In pairwise comparisons, these cancer genes showed distinct patterns of co-mutation and mutual exclusivity both with each other and with recurrent cytogenetic abnormalities (Supplementary Fig. 2j), as previously described[9,13]. However, a pairwise approach does not have the power to detect tertiary, quaternary and higher-level interactions which are expected in a heterogeneous disease as MM[9]. Therefore, higher-level statistics must be used to deconvolute the complex landscape of MM and to identify subgroups of cases with a similar genomic landscape. To this end, we developed a composite analysis employing both Bayesian networks and the hierarchical Dirichlet process (hdp)[23] to define the conditional dependencies of all driver events in MM and attempt a genomics-based classification of the disease (Fig. 1b, c, Supplementary Fig. 3 and Supplementary Software 1)[23,24]. Bayesian network (BN) analysis identified both known and novel patterns of higher-level co-occurrence and mutually exclusivity between driver events. In particular, we observed a pattern of co-occurrence between t(4;14)(MMSET; IGH), TRAF3 deletion and 13q14 deletion (Supplementary Fig. 3a). Furthermore, together with well-known mutually exclusive patterns, such as the one between IGH translocations and hyperdiploid cases (Supplementary Fig. 3b), several novel ones were identified. Specifically, FAM46C and CDKN2C were frequently deleted together, and when this happens the co-deletion was mutually exclusive with t(4;14)(MMSET;IGH) (Supplementary Fig. 3c); TRAF3 deletions were associated with NFKBIA and showed a mutually exclusive pattern with t(11;14) (CCND1;IGH) and t(14;16)(IGH;MAF) (Supplementary Fig. 3b, d); MAX mutations were usually not associated with hyperdiploid cytogenetic status (Supplementary Fig. 3e). Finally, CYLD deletions were associated with both HRD and t(11;14)(CCND1;IGH) and mutually exclusive with t(4;14)(MMSET;IGH) (Supplementary Fig. 3f).

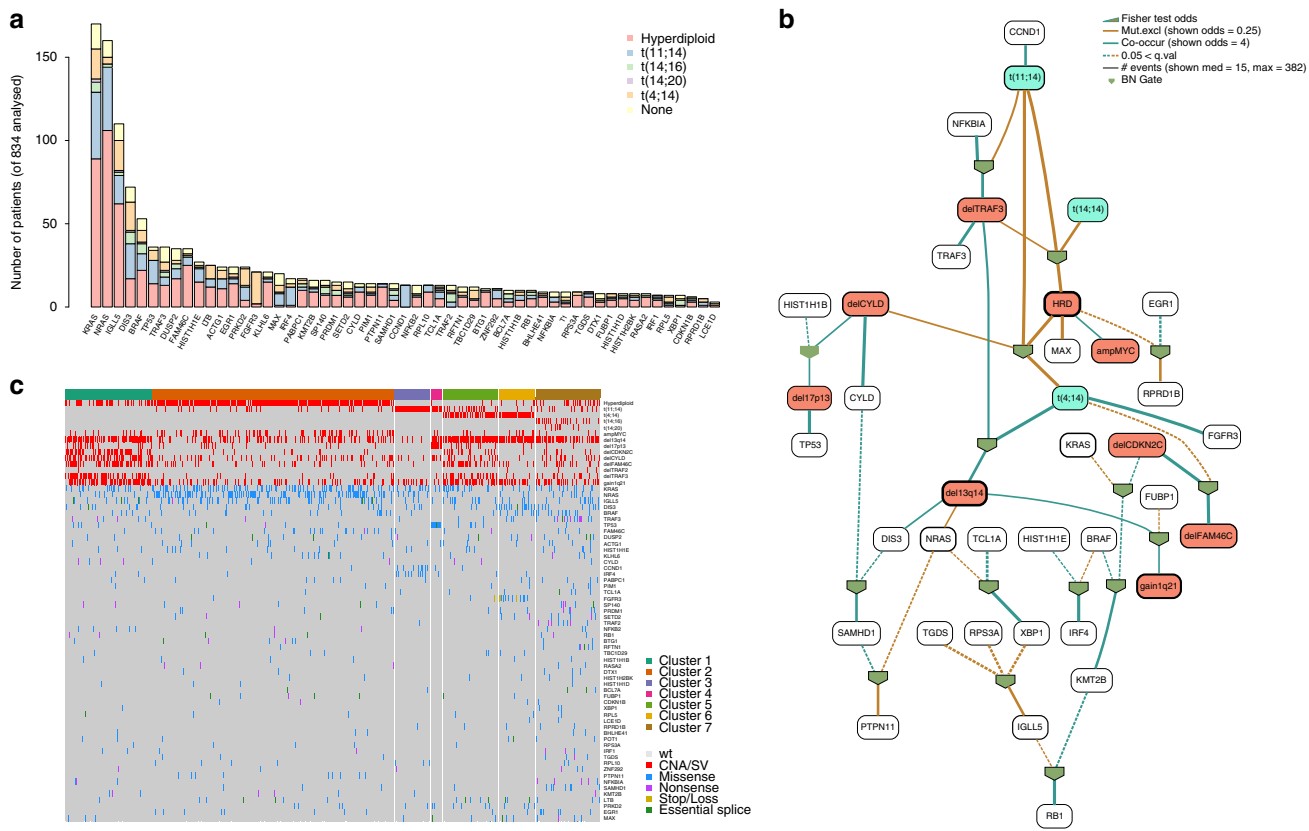

**Fig. 1** The landscape of exome-based driver events MM. **a** Landscape of driver mutations in MM. Each bar represents a distinct positively selected gene and each bar's color indicates its prevalence across the main MM cytogenetic subgroups. **b** We built the optimal Bayesian network by considering the recurrent SVs and CNAs (n = 14) and driver SNVs (n = 55) across 724 MM patients, where the final list of 69 variables was assessed. To further investigate the type of recurrence patterns we fitted logic gates between parent and child nodes in the network. The gate combination with the most significant Fisher exact test p value was selected. The line width is proportional to the log hazard ratio of the test. Dashed lines represent non-significant associations (p > 0.05). CNAs and translocations were colored in brown and light blue respectively. The thickness of the outline of each box is proportional to the prevalence of the event across the entire series. **c** The heatmap showing the main MM genomic subgroups across 724 MM patients. The genomic profile of each cluster was generated by integrating the hierarchical Dirichlet process and Bayesian network data. Rows in the graph represent individual genomic lesions, and the columns represent patients

The BN approach identified several nodes of this complex network of interactions, mainly represented by cytogenetic lesions. We shortlisted these and added them to the list of driver gene mutations above to extract clusters of MM cases with a similar landscape of driver genomic events (Supplementary Software 1 and "Methods"). Different from previous approaches[9,13], our analysis included the full catalog of point mutations in MM driver genes (defined by dNdScv), recurrent SVs and CNAs, representing a final list of 69 driver events. We identified 7 groups (Fig. 1c, Supplementary Software 1), where the strongest determinants of genomic substructure were *IGH* translocations and recurrent CNAs. Some co-operating genetic lesions were nonrandomly distributed across these main groups: a significant fraction of patients without *IGH* translocations were generally enriched for 1q gain and deletions on 1p13, 1p32, 13q, *TRAF3*, and *CYLD* deletions (*Cluster 1*). RAS signaling mutations, especially *NRAS* and *KRAS*, were associated with hyperdiploidy and *MYC* amplification (*Cluster 2*). A significant fraction (33%) of patients with t(11;14)(*CCND1*;*IGH*) were characterized by low genomic complexity and high prevalence of *IRF4* and *CCDN1* mutations (*Cluster 3*). Patients harboring *TP53* bi-allelic inactivation were clustered in an independent sub group (*Cluster 4*) and displayed by far the worst prognosis as also recently demonstrated (Supplementary Software 1)[13]. A significant fraction of t (4;14)(*MMSET*;*IGH*) (51%) and t(11;14)(*CCND1*;*IGH*) (19%)

patients was characterized by multiple cytogenetic aberrations, with low prevalence of *CYLD/FAM46C* and *TRAF3* deletions, respectively (*Cluster 5*). A second fraction of patients with t(4;14) (*MMSET*;*IGH*) translocation (46%) were grouped with deletion of 13q14, gain of 1q21, *DIS3*, and *FGFR3* mutations (*Cluster 6*). Finally, patients without hyperdiploidy or the common *IGH* translocations were characterized by a generally higher number of driver gene mutations, across a wide range of myeloma cancer genes (*Cluster 7*). Included in this final cluster was patients with t (14;16) and t(14;20) translocations, between *IGH* and c-*MAF* or *MAFB*, respectively, it may be that the small numbers of patients with these events make it difficult to identify them as a separate cluster.

**Patterns of structural variation in myeloma.** Aside from the few recurrent oncogenic translocations and aneuploidies, we lack an unbiased and comprehensive description of genome-wide patterns of SVs in MM. In the WGS data, we identified 2122 SVs, with a heterogeneous distribution across the cohort (median 26 per patients, range 0–129) (Fig. 2a), suggesting that structural variation is a major force shaping the MM genome. Translocations involving known targets such as the canonical *IGH*-oncogene and *MYC* translocations only accounted for 6.5% (137/2122) of all SVs (Supplementary Fig. 4a, b). Instead, most SVs were

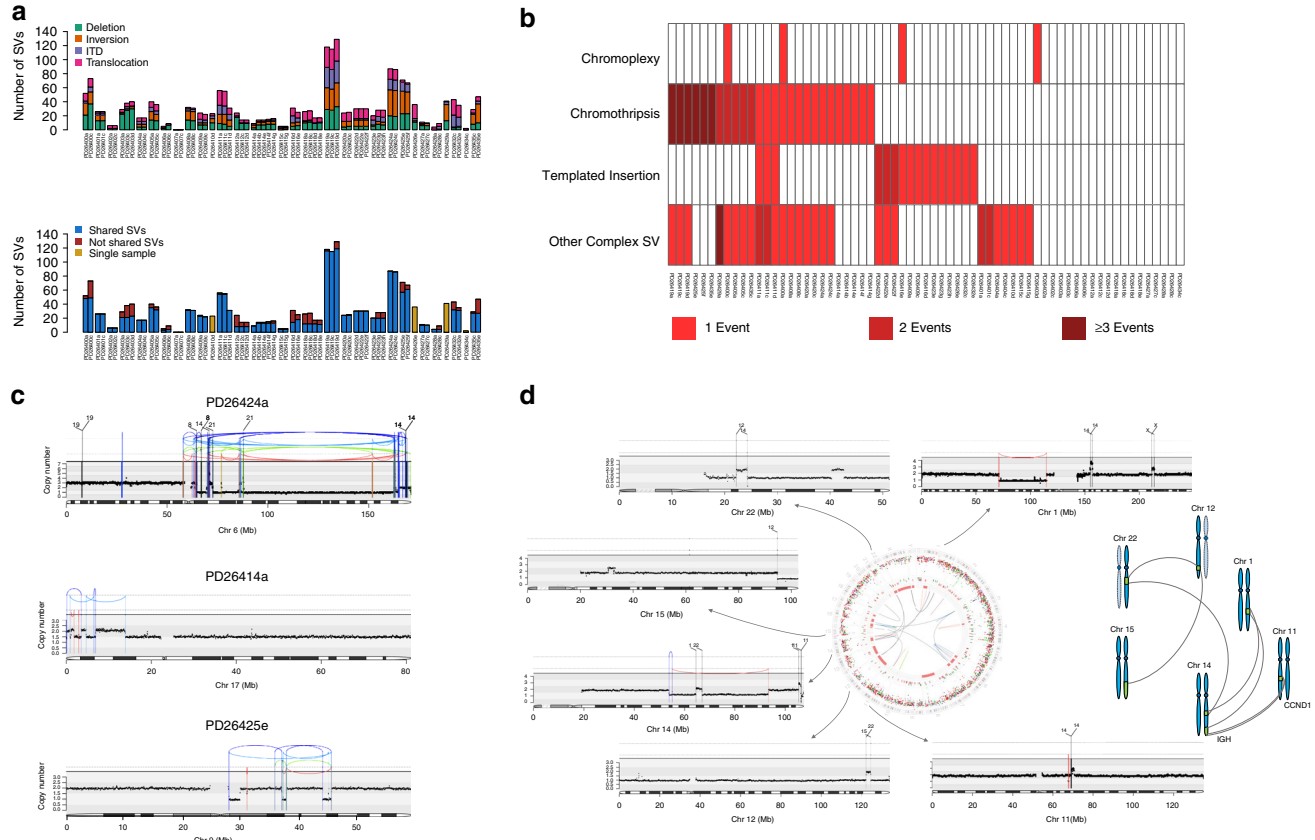

**Fig. 2** The landscape of structural variants (SVs) in MM. **a** Top, prevalence of SVs across the entire series. Bottom, the proportion of SVs shared between samples collected at different time points within the same patients. **b** A heatmap representing the distribution and prevalence of the main complex events: chromothripsis, chromoplexy, and templated insertions. **c** Three examples of chromothripsis. In these plots, the red arch represents a deletion, the green arch represents an internal tandem duplication (ITD) and the blue arch represents an inversion. **d** Example of templated insertion. In the middle, the genome plot of patient PD26422 represents all main genomic events: mutations (external circle), indels (middle circle; dark green and red lines represent insertions and deletions respectively), copy-number variants (red = deletions, green = gain) and rearrangements (blue = inversions, red = deletionss, green = ITDs, black = translocations). Externally, a copy-number/rearrangement plot of each chromosome involved in the templated insertion is provided, highlighting a focal CNA around each breakpoint. This case represents a clear example of how templated insertion may involve critical driver oncogenes, like *CCND1* in this case. A schematic representation of this sample templated insertion is reported on the right

private and included many unbalanced translocations and complex events (Supplementary Fig. 4c–f, "Methods")[25]. Most (24/30; 80%) patients had at least one complex SV ("Methods"), of which chromothripsis was the most frequent (11/30; 36%) (Fig. 2b, c)[25–29]. Chromoplexy occurred in three patients (Fig. 2b, Supplementary Fig. 5a–c)[25].

Interestingly, we also identified a novel pattern of complex SVs, recently observed in some solid cancers[25], characterized by cycles of templated insertions, found in 6/30 (20%) patients. Here, several low-amplitude copy-number gains on different chromosomes were linked together through SVs demarcating the region of duplication (Fig. 2b, d; Supplementary Fig. 5d–h). The presence of multiple copy-number gains suggests a replication-based event, where the most plausible explanation for this pattern is that the templates are strung together into a single chain, hosted within one of the chromosomes.

Known driver genes were common targets of complex SV events, including *MYC* (13/30 cases; 43%), *CCND1* (8/30; 26%), and *MMSET* (3/30; 10%). For example, the juxtaposition of *CCND1* to the *IGH* locus was caused by either unbalanced translocations or insertional events in 5/8 patients (Supplementary Fig. 6). Similarly, *MYC* translocations showed unanticipated complexity, with four cases of cycles of templated insertions, one chromoplexy and one chromothripsis involving *MYC* or its

regulatory regions (Supplementary Fig. 7). Such events are the structural basis of the oncogene amplification observed by FISH in many cases of t(11;14) and t(8;14)[30,31]. Interestingly, many of the *MYC* SVs involved the immunoglobulin light chain loci, *IGK* or *IGL*, rather than the heavy chain *IGH* locus, and were seen in patients with hyperdiploidy (Supplementary Fig. 4b). Although sometimes occurring late, these events were under strong selective pressure: we identified a striking case of convergent evolution where a subclone bearing an *IGL:MYC* translocation was lost and one bearing an *IGH:MYC* was selected at relapse (Supplementary Fig. 8). SVs also led to loss of tumor suppressor genes such as *BIRC2/3*, *CDKN2A/B*, *CDKN2C*, *TRAF3*, and *FAM46C*[32], either within focal deletions or more complex events. These data confirm that structural variation, accessing both simple and complex mechanisms of genome rearrangement, is an important process driving MM evolution.

**Timing of aneuploidies and SVs.** Genomic studies have started to investigate the temporal windows of occurrence of events in MM, mainly through the identification of subclonal (late) point mutations and SV appearing or disappearing in serial samples[5,10,12,33,34]. In our cohort of serial WGS samples, we could add significant resolution to this analysis through integration of SVs, CNAs, and point mutations. This investigation was

particularly relevant for hyperdiploid patients (18/30). In fact, even though trisomies of odd chromosomes are one of the hallmark lesions of myeloma[1–3], their mode of acquisition is still a matter of debate. If gains occur as independent events, subclonal evolution within the myeloma cells between diagnosis and relapse could lead to considerable diversification of chromosome complements. Consistent with this, we observed significant karyotype changes within the same patient over time, including loss or acquisition of some gains in hyperdiploid patients (Supplementary Fig. 9)[34]. At the extreme end of this cytogenetic dynamism, we found 4/30 patients acquiring a whole-genome duplication at relapse, highlighting how multiple gains events may play a significant role in relapsed/refractory stages and copy-number gains may be acquired in different time windows (Fig. 3a, b and Supplementary Fig. 9).

Our data thus suggest that even seemingly clonal CNAs could actually be acquired late in a fraction of cells, but this could only

be revealed by analysis of serial samples. We therefore sought to focus on the relative order of acquisition of clonal chromosomal gains in single samples through a molecular time analysis based on the fraction of duplicated to single-copy clonal (early) point mutations[35]. The concept underpinning this approach is that any trisomy should carry a high number of duplicated clonal mutations if the gain occurred late in MM evolution, but only few duplicated mutations in early gains ("Methods"). Indeed, we found that the hallmark trisomies of hyperdiploidy tended to be present in the ancestral clone of the myeloma, but within a given patient were not always acquired simultaneously (Fig. 3c–e and Supplementary Fig. 10) and rather showed an heterogenous pattern of accumulation (Fig. 4a). To validate this approach, we applied our molecular time analysis to serial samples. There, apparently clonal CNAs on a single sample that were nevertheless unstable across the series were correctly assigned to a later time window (Supplementary Fig. 11).

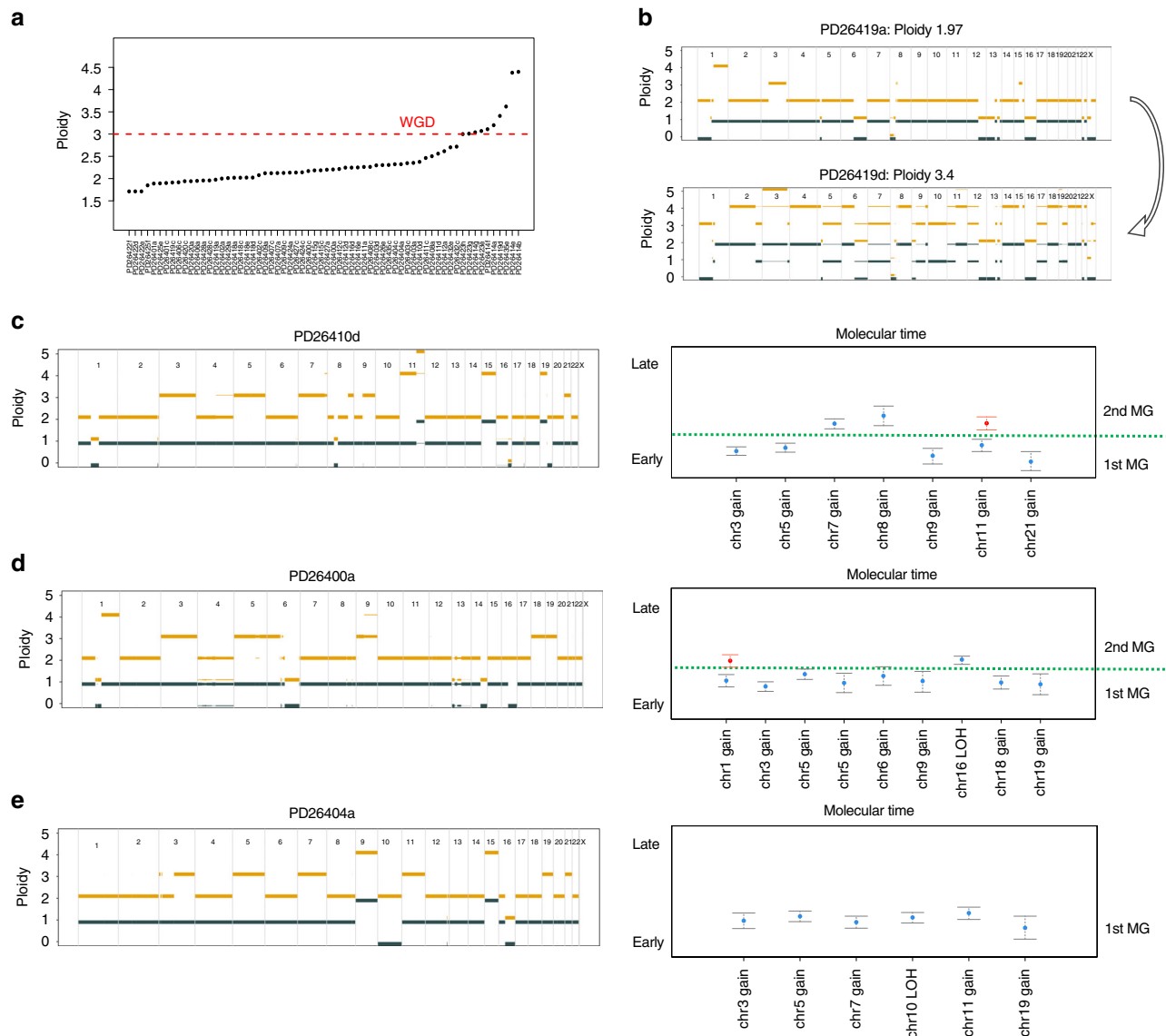

**Fig. 3** Timing the clonal number changes in MM. **a** Summary of the sample's ploidy for the entire series. Samples with ploidy > 3 (above the dashed red line) were considered as whole-genome duplication (WGD). **b** The copy-number profile of a MM patient (PD26419) that experienced a WGD at relapse: gold line = total copy number, gray = copy number of the minor allele. The presence of more than 1 cytogenetic segment is compatible with the existence of a subclonal CNA whose CCF is proportional to the segment thickness. **c–e** Left, standard copy number profile of 3 hyperdiploid MMs. Right, the molecular time (blue dots) estimated for each clonal gain and copy-neutral loss of heterozygosity ("Methods"). Red dots represent the molecular time of a second extra gain occurred on a previous one. Dashed green lines separate multi gain events occurring at different time windows

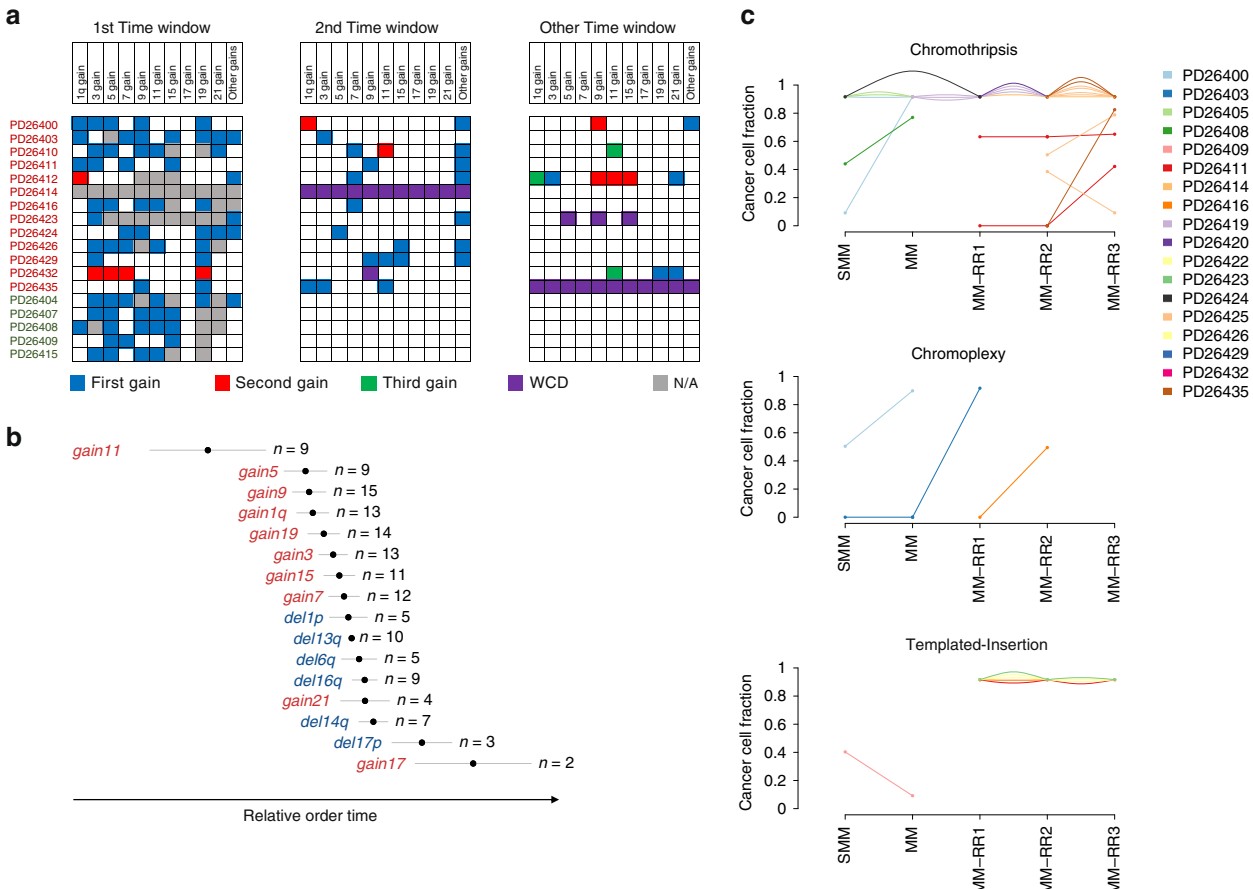

**Fig. 4** The chronological reconstruction of aneuploidies acquisition in MM. **a** Heatmaps representing the cumulative acquisition of copy-number gains observed in 13/18 hyperdiploid patients, labeled in red if the final HRD profile was generated by multiple and independent events ($n = 13$) or green if the trisomies were acquired in one single time window ($n = 5$). Boxes are color-coded based on the relative order of acquisition of each event; WCD = whole chromosome duplication. **b** A Bradley–Terry model based on the integration between the CCF and molecular time of each recurrent CNAs (gains = red and deletions = blue) for all 30 MM cases included in this study. Segments were ordered from the earliest (top) to the latest (bottom) occurring in relative time from sampling. The time scale (X-axis) is relative since timing of genomic evolution is variable from case to case and not easily correlated to age. **c** Cancer cell fraction of each single complex event for each patient over the time

We could also time multiple duplications of the same chromosome (Fig. 3c–e and "Methods"). Interestingly we observed an heterogenous pattern of accumulation (Fig. 3c–e and Fig. 4a). 1q gains were recurrently involved in multiple duplications (7/14 cases), and we observed that in all but one patient the second or third 1q gain was acquired in an independent and late time window; in contrast the first 1q gain was generally acquired together with other trisomies in the earliest time window (Supplementary Fig. 10). Likely, this provides the biological basis for the adverse prognostic effect of multiple-copy gain of chr1q (a late event associated with progression) but not of single gain (an early event potentially associated with initiation)[36].

Combining data from molecular time analysis and the clonality/stability of the trisomies over time, we reconstructed the timing of acquisition of aneuploidies in each patient and analyzed trends in the whole cohort. We found that in 13/18 hyperdiploid patients gains were acquired in different and independent time windows (Fig. 4a). Overall, gains of odd chromosomes and the first 1q gain were amongst the earliest in our series, and recurrent chromosome losses were acquired later than trisomies (Fig. 4b). These data are consistent with the proposition that hyperdiploidy is an early driver event[1,3], and suggests a potential early role of a single 1q gain in MM pathogenesis.

Translocations involving *CCND1* and *MMSET* were always fully clonal, confirming their early driver role in MM pathogenesis. *MYC* translocations showed a more heterogenous pattern, with 8/13 events clonal and stable over time. Interestingly, this was biased toward newly diagnosed or relapsed MM, where this recurrent translocation was clonal and conserved between different samples in 7/9 patients. Conversely, *MYC* aberrations were detected at subclonal level in 3/4 samples collected at the smoldering MM stage, confirming preliminary data that support a role of *MYC* dysregulation during progression to symptomatic MM[33,37]. Chromothripsis and cycles of templated insertion events were mostly clonal and conserved during evolution (17/24 and 5/6, respectively), suggesting they occurred early in MM pathogenesis (Fig. 4c). However, a fraction of patients showed some evidence of late chromothripsis (7/24), implying a possible involvement in late cancer progression (Fig. 4c and Supplementary Fig. 9d). Conversely, chromoplexy emerged as a late event being positively selected and/or acquired at progression of smoldering into symptomatic MM (one patient) or at relapse (two patients) (Fig. 4c).

**Preferred evolutionary trajectories of myeloma development.** We integrated all extracted chronological data on SVs, hyperdiploidy and point mutations to generate phylogenetic trees for each sample ("Methods", examples in Supplementary Fig. 12,

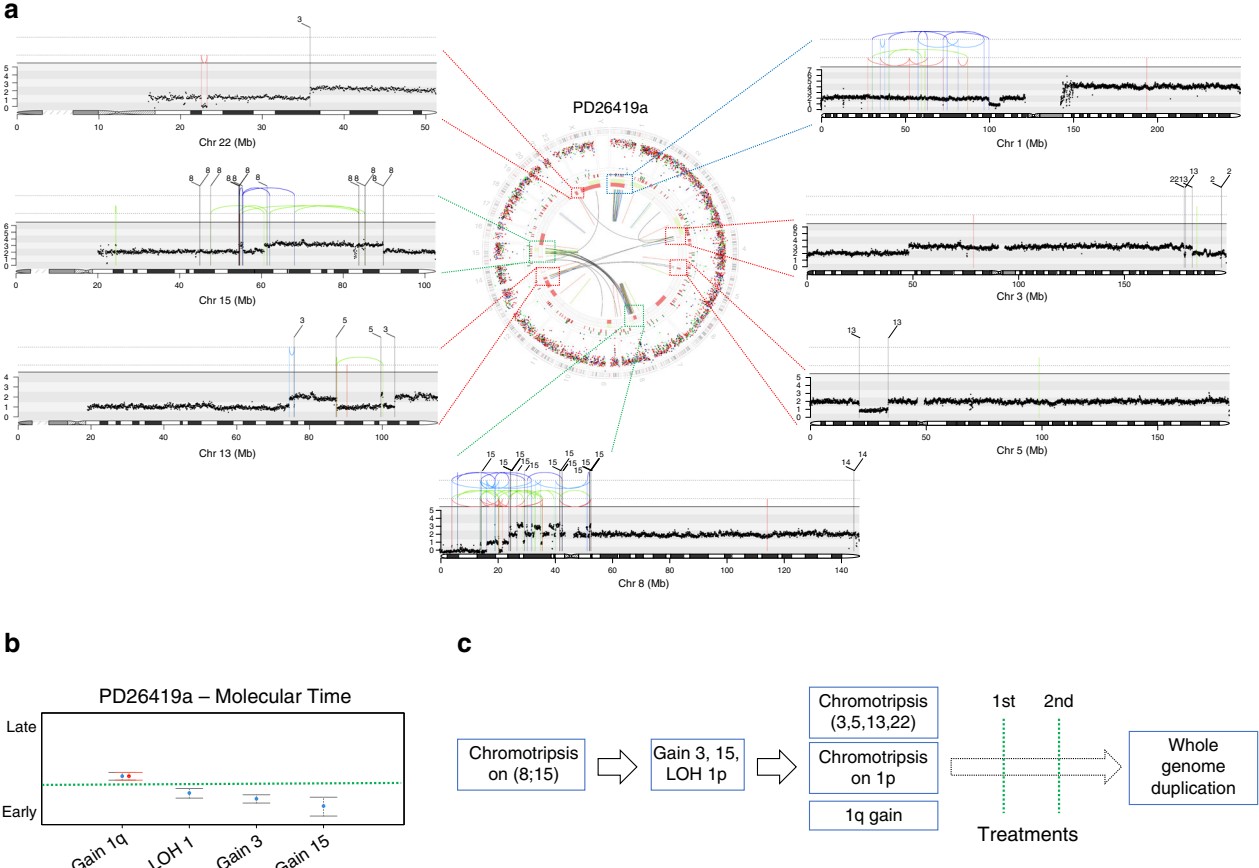

**Fig. 5** Defining the relative order of acquisition of the key driver events. **a** Genome plot of patient PD26419a where the three chromothripsis events [(8;15), (3;5;13;22) and 1p] were highlighted with different colored dashed lines connected to specific rearrangements/copy-number plots. In these plots, the red arch represents a deletion, the green arch represents an ITD and the blue arch represents an inversion. **b** Molecular time of the main clonal gains and CN-LOH in the PD26419a sample. This data suggested the existence of at least two independent time windows: the first involving alterations on chromosomes 3, 15, and 1p and the second chromosome 1q. **c** The driver events of patient PD26419 are reconstructed in chronological order

Supplementary Data 4)[5,38]. Differently from previous studies, this analysis was not just limited to the differentiation between clonal (early) and subclonal (late) events, but inferred molecular time data from other genomic events (i.e., SVs, CNAs, and driver mutations), allowing assessment of the order of acquisition of early clonal driver events in MM in preclinical phases ("Methods"). Our methodology is worked through for one illustrative patient carrying (i) several clonal chromosomal gains, (ii) 3 distinct and clonal chromothripsis events, and (iii) a whole-genome duplication (Fig. 5). One chromothripsis involved chromosomes 8 and 15, duplicating the long arm of chromosome 15. Because few mutations were present on chromosome 15 at the time it duplicated, this must have occurred early in molecular time (Fig. 5b). Gain of chromosome 3 and copy-neutral loss-of-heterozygosity (CN-LOH) of small arm of chromosome 1 (chr1p) occurred in the chromosome 15 gain's time window, and they were followed by a second chromosomal crisis involving chromosomes 3, 5, 13, and 22. This chromothripsis must have occurred on one of the two duplicated alleles of chromosome 3 (and therefore after the acquisition of a chromosome 3 trisomy) because regions of copy-number loss within the chromothripsis had a copy number of 2 and SNPs were heterozygous. Within the same time window, two more events ensued: a new chromothripsis on chr1p after CN-LOH, and an amplification of 1q to a CN status of 4. Finally, this patient underwent whole-genome duplication after two therapy lines, as highlighted by samples from relapsed-refractory stages (Fig. 3b).

Through similar analyses, we were able to chronologically reconstruct the order of acquisition of early and late (sub)clonal events in all patients, potentially looking at timing of events acquired years before sampling. The trunks of the phylogenetic trees of 29/30 (97%) patients were characterized by few genomic events, generally acquired during different time windows of the MM life history leading to the emergence of the most recent common ancestor (Fig. 6). These events were acquired with a nonrandom order. Overall, chromothripsis, cycles of templated insertions, chromosomal gains and other SVs accounted for most of the earliest events, emerging as key drivers of disease initiation, seeding the soil for driver events that would later arise and confer further selective advantage to the clone. Focal deletions on distinct oncogenes, whole-genome duplication and chromoplexy were generally acquired during progression and/or relapse, potentially representing new mechanisms of subclonal selection and treatment resistance[5,10].

WGS thus offered a more detailed compendium of driver events, mostly structural, and of their chronological order of acquisition. We next explored how this enriched view of the genome offered by WGS fitted within the groups identified by the combined hdp and BN clustering, based on whole-exome sequencing of the CoMMpass dataset (Fig. 1c). Samples assigned to Clusters 1 and 4 were characterized by multiple and independent complex events acquired during different time windows, in line with their complex driver profile observed in the combined clustering. Patients assigned to Cluster 3 [low

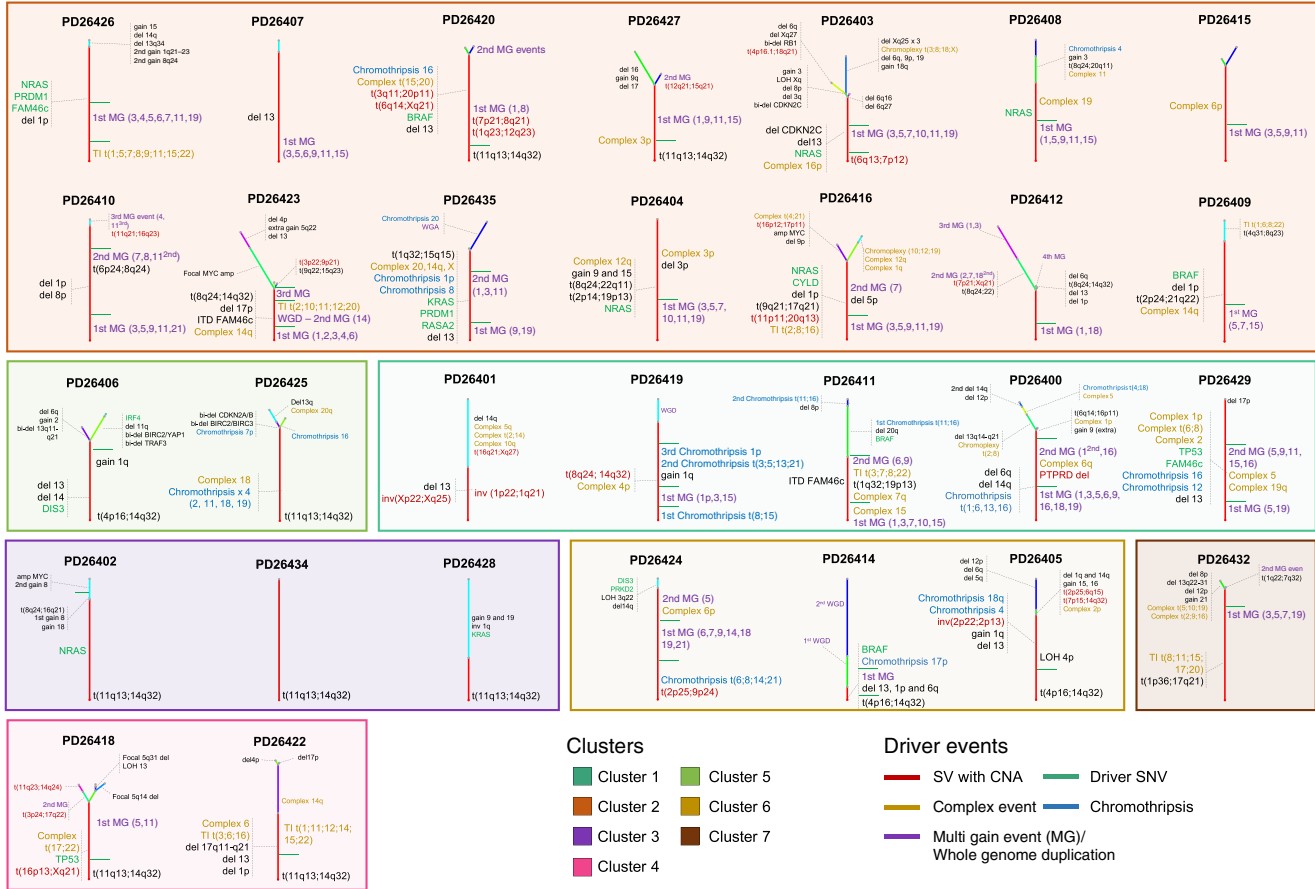

**Fig. 6** The chronological reconstruction of driver events in MM. Phylogenetic trees generated from the Dirichlet process analysis ("Methods"). The root (always colored in red) and branch length are proportional to the (sub)clone mutational load. All main drivers (CNAs, SNVs, and SVs) were annotated according to their chronological occurrence. Early clonal events (root) were chronologically annotated on the right, when it was possible to establish a specific time window. Here, all different root time windows were separated by a green line. Conversely, early drivers without a clear timing were grouped together on the left of the root. All driver events that occurred in the root were reported with larger font size. Patients were grouped according to the genomic clustering shown in Fig. 1c, and color-coded accordingly. TI = templated insertion

genomic impairment and t(11;14)(CCND1;IGH)] showed a high grade of similarities also at WGS level, where we identified in all cases few genomic lesions and the absence of both multigain and complex events. In contrast, cases assigned to Clusters 2, 5, and 6 showed variable levels of genomic complexity. Overall this suggests that, although exome/targeted approaches are able to characterize mutations, recurrent CNAs and oncogene translocations, they are not completely able to fully decipher the MM genomic complexity, and that WGS on a larger number of samples will yield novel insights in MM pathogenesis.

## Discussion

Our study provides several novel insights into MM biology, taking advantage of WGS and serial sampling. For the first time, we comprehensively describe the landscape of SVs and complex events in MM, showing how these genomic aberrations are pivotal for MM pathogenesis, but play differential roles during different evolution phases. Interestingly, complex events such as chromothripsis and cycles of templated insertions are strongly players in MM, with an unexpectedly high prevalence that could not be captured by previous exome-based studies. Interestingly, these complex events had a key role in early phases of cancer development. Conversely, other events, such as focal deletions on oncogenes, chromoplexy and WGD, emerged as later aberrations potentially involved in drug resistance and relapse.

We demonstrate here how hyperdiploidy reflects the sum of multiple and independent chromosomal gains over time, in contrast to prior hypotheses[1–3]. This early and multi-step acquisition of trisomies is in line to what has been recently reported in solid tumors in a large pan-cancer analysis[35]. However, in contrast to solid cancers[35,39,40], MM does not show a reliable relationship between global mutation burden and age, probably because of mutational processes that are not linear in time, and rather act in bursts, such as those promoted by AID and APOBEC[33]. Therefore, although we can be confident that the first trisomies occur early in relative molecular time, we cannot provide robust estimates of the chronological age at which these events were acquired.

The combined chronological reconstruction of SNVs, CNAs, and SVs suggests that MM development follows preferred evolutionary trajectories, with stuttering accumulation of driver events in keeping with its insidiously progressive but unpredictable clinical course. Critical early events include immunoglobulin translocations with MMSET and CCND1, hyperdiploidy and complex structural variation processes hitting key myeloma genes. These early driver mutations shape the subsequent evolution of myeloma, each with preferred sets of co-operating cancer genes. The majority of these events are only detectable by WGS, highlighting a significant limitation of exome/targeted sequencing approaches to fully decipher the complexity of MM. In the near future, large cohorts of MM studied by WGS will

better characterize these evolutionary trajectories over time, opening the field for more rationale therapeutic and preventive strategies.

## Methods

**Sample selection**. The study involved the use of human samples, which were collected after written informed consent was obtained (Wellcome Sanger Institute protocol number 15/046). Samples and data were obtained and managed in accordance with the Declaration of Helsinki. DNA was extracted from CD138+ cells purified from bone marrow from 30 patients, of which 26 (86%) with multiple sampling, for a total of 67 tumor samples and 30 matched normal samples (Supplementary Data 1). Samples were collected at different clinical time points: smoldering (n = 11), symptomatic (n = 15), and relapsed MM (n = 41) (Supplementary Data 2). Constitutional control DNA originated from peripheral blood mononuclear cells. Purity of the CD138+ fraction was assessed by anti-CD138 immunocytochemistry post sorting, and only samples with >90% plasma cells were sequenced.

**Massively parallel sequencing and alignment**. Short insert 500 bp genomic libraries were constructed, flowcells prepared and sequencing clusters generated according to Illumina protocols. We performed 108 base/100 base (genomic) paired-end sequencing on HiSeq X10 genome analyzers. The average sequence coverage was 38.7-fold. Short insert paired-end reads were aligned to the reference human genome (GRCh37) using the Burrows–Wheeler Aligner (BWA) (v0.5.9)[41].

**Processing of genomic data**. CaVEMan (Cancer Variants Through Expectation Maximization: http://cancerit.github.io/CaVEMan/) was used to call somatic substitutions[42,43]. Indels were called using a modified Pindel version 2.0 (http://cancerit.github.io/cgpPindel/) on the NCBI37 genome build[44]. SVs were discovered using a bespoke algorithm, BRASS (Breakpoint Analysis) (https://github.com/cancerit/BRASS) analyzing discordantly mapping paired-end reads. Discordantly mapping read pairs that were likely to span breakpoints, as well as a selection of nearby properly paired reads, were grouped for each region of interest[43]. In patients with multiple samples, all discordant SVs were evaluated by PCR and by manual curation on Integrative Genomics Viewer[45] in order to confirm the effective acquisition or loss of the rearrangement. Chromotripsis and chromoplexy were defined as previously described[26,27]. We defined as complex events the following SV classes: chromothripsis, templated insertions among three or more chromosomes, chromoplexy and all other complex events composed by more than 3 SVs, as recently described[25].

Allele-specific copy-number analysis of tumors was performed by applying ASCAT (v2.1.1), to next-generation sequencing data[46]. The evaluation of copy-number changes, also including subclonal and minor aberrations, was performed by Battenberg process[38,47]. Both ASCAT and Battenberg were used to correctly estimate the cancer cell fraction (CCF) for each sample.

**Driver mutations**. To identify genes under positive selection in MM, i.e., drivers of tumor progression, we relied on the recently published dNdScv method, which estimates the excess of nonsynonymous mutations while accounting for the mutational spectrum and gene-specific mutation rates[16,17]. dNdScv was run on a series of 834 MM patients: 30 patients from this cohort and 804 patients from the CoMMpass series. The CoMMpass data were generated as part of the Multiple Myeloma Research Foundation Personalized Medicine Initiatives (https://research.themmrf.org and www.themmrf.org). We run dNdScv on the entire gene set and, to increase statistical power, on a restricted set including recently reported potential MM driver genes[13]. In addition to identifying positively selected genes, dNdScv estimates the excess in nonsynonymous mutations across a set of genes, making it possible to calculate the average number of driver mutations per sample, as recently reported[17].

**BN analysis**. The BN was constructed by first creating a binary matrix of the presence/absence of genomic aberrations and mutations in each patient. The Gobnilp[48] software was then used with default parameters. These include a maximum of three parents. The algorithm finds the globally optimal BN within the constraints set, by posing the learning of the BN structure as a linear optimization problem that is solved using integer linear programming via the SCIP Optimization suite[49]. For the resulting BN, we wrote bespoke functions in SWI-Prolog[50] that find the logic gates combination which maximizes the Fisher exact test P value when regressing the parents against the child (fisher.test() R function)[51]. Gate formulae are constructed by considering all possible gate combinations that involve the parental edges to a specific child node. From all the formulae we selected the one that maximizes the p value of the Fisher's exact test and uses the least number of gates. Gate vectors are produced by performing the gate operation on its input vectors. The output vector can then be tested against the child vector. Logic gates in sequence are abbreviated, thus in the presented BN the two AND gates: [t(4;14) AND HRD AND t(14;16)] are abbreviated to a single AND gate with three inputs: AND[t(4;14),t(14;16),t(11;14)].

The BN allows the identification of both complex and simple pattern correlations of drivers. The output is composed by two different figures: (i) the global BN figure (Fig. 1b), where all the gates/correlations are listed and connected; (ii) the heatmap of driver events for each family in the BN, which allow easy interpretation by scientist through visual inspection (Supplementary Figs. 3 and 13).

**Hierarchical Dirichlet process**. The hdp was used to investigate the main MM genomic subgroups (https://github.com/nicolaroberts/hdp)[23]. All dN/dS-extracted driver mutations and the most clinically relevant and recurrent CNAs and SVs were included, for a total of 69 variables available for 724 (87%) cases.

The final exome-based clusters were extracted combining the hdp and BN data (Supplementary Figs. 14 and 15).

The full analysis process written in R is provided in Supplementary Software 1.

**Construction of subclone phylogeny**. Somatic mutations were clustered using a Bayesian clustering method[5,38]. This method assumes a mixture model for the counts of mutant and wild-type reads in a series of NGS read samples, and a Dirichlet process prior on the sequence of category (or cluster) weights. In this model, the count of reads carrying a particular mutation, in the reads from a specific tissue sample, follows a binomial distribution (conditional on the assignment of the mutation to a subclone, and on the frequency of this subclone among the tumor cells from the specified tissue sample). A Markov chain Monte Carlo (MCMC) sampler (based on Gibbs sampling) was used to sample from the posterior distribution of the subclone (cluster) weights, the subclone (cluster) assignments of the mutations, and the frequency of each subclone among the tumor cells from the specified tissue sample. The MCMC sampler was run for 1000 iterations, of which the first 300 were discarded. The MCMC output was post-processed, as described in Nik-Zainal et al. and Bolli et al.[5,38], to obtain point estimates of: (i) the subclone (or cluster) weights; (ii) the assignment of mutations to subclones; (iii) the frequency of each subclone among the tumor cells from each tissue sample.

In order to identify the phylogenetic relationships between subclones, we followed the previously published approach[5,38], and this version of the "pigeonhole principle": if the proportion of cells carrying mutation A is p_A, and the proportion carrying mutation B is p_B, and p_A + p_B > 1, then at least one cell must carry both mutations A and B. We also assume that within a tumor, each mutation occurred as a unique event. So, p_A + p_B > 1 and p_B < p_A implies that all cells which carry mutation B must also carry mutation A. We can represent the relationships among subclones as a phylogenetic tree, in which each node represents a subclone. We can also interpret the branch which ends in a node as representing the same subclones as this node (it is more natural to think of branches as representing subclones in this way, as mutations occurred in some temporal order along each branch of the tree). For any proposed set of parent–offspring relationships among a set of subclones, we can check if the above pigeonhole principle is respected by the subclone frequency parameters for every tissue sample. If the pigeonhole principle is respected for every tissue sample, then we say that this set of parent–offspring relationships is "allowed". In practice, we relaxed this condition slightly, by introducing a "tolerated error" parameter, to which we have assigned the value 0.001. This means that when comparing the subclone frequencies of a subclone with its putative daughter subclones, we allow the total frequency of the daughter subclones to exceed the frequency of the parental subclone by as much as 0.001 before we declare a violation of the pigeonhole principle. We have also introduced an "acceptable mutant count" parameter, to which we have assigned the value 50. This means that all subclones which contain fewer than 50 mutations were excluded.

**Tree finding algorithm**. We describe below a simple algorithm to identify all trees which are allowed (compatible with the pigeonhole principle) by the subclone frequency parameters for every tissue sample. We have implemented this algorithm as a function in the R language (Supplementary Software 2). This R function also plots the trees. In these tree plots, the length of each tree branch is proportional to the number of mutations assigned to the corresponding subclone.

Let us denote the list of k subclones s = (1, 2,..., k). We can represent a tree as a list of k elements t = (t[1], t[2],...,t[k]), in which t[i] denotes the ancestor of subclone i. The element t[i] can take a value from the list of subclones s, or the value 0 (to indicate that subclone i has not been assigned an ancestor). We will also use the notation s/a to denote a list which contains every element of list except element a which has been deleted.

In the step 1, it identifies all those subclones in list s which could be the root node of the tree, and place these possible root nodes in a new list r. For each candidate root node, take every other subclone, and test whether this subclone is allowed to be a daughter node of the candidate root node. Only if every other subclone is an allowed daughter node of the candidate root node, can we accept the candidate as a possible root node. In many cases, there is only one such node.

In the step 2 it creates an empty list of trees. Notice that in general, the list p(i, t) of possible parent nodes of subclone i in tree t does depend on the current tree t. This is because subclone i may already have been assigned daughter nodes in tree t.

These daughter nodes, and any other descendant nodes of subclone i, must be excluded from the list of possible parent nodes of subclone i.

The resulting list T of trees may be empty, indicating that we cannot construct a tree which is compatible with the pigeonhole principle, given the subclone frequency parameters for every tissue sample. In practice, the list T often contains only a single tree.

Supplementary Fig. 16 shows two examples of the phylogenetic three generation from the Dirichlet clustering data (2D-plot)[5,33].

**Molecular time and copy-number timing.** During a copy-number gain, all SNVs already acquired on the involved allele will be duplicated as well[52]. Consequently, the variant allelic frequency corrected for the nontumor cells contamination (c-VAF) of all these clonal duplicated SNVs will change from 50% to ~66%, being present on two out of three alleles. Consequently, all clonal SNVs occurred (i) anytime on the nonduplicated allele or (ii) on one of the two duplicated alleles after the duplication will have a c-VAF ~33% and all subclonal SNVs will always occur on one single allele with a c-VAF <33%. (Supplementary Fig. 17a)

To assign each SNV to either a pregain or postgain status, we have created a script that uses *mclust* R function for this analysis. Our R function is called "*mol_time*" and is available on github.com/nicos-angelopoulos/mol_time. The input to the function is the corrected c-VAF profiles as shown in Supplementary Fig. 17. In panels (b) and (c) *mclust* was run on 2 chromosome gains dividing the SNV catalog in two main groups: one where mutations were detected on two alleles and one where mutations were observed only of one allele out of three. In panel (d) *mclust* was run on a CN-LOH, and this explains why duplicated and nonduplicated mutations had 100% and 50% c-VAF, respectively. Finally, in the last plot (e) differently from the recently published molecular time analysis[35], *mclust* was also run in the presence of two extra copies of the same chromosome. The fact that we have mutations present on three and on two chromosomes suggested that these two gains were acquired at different time points. If we had only one group of mutations at 75% c-VAF (tripled SNVs), it would have been compatible with a tetrasomy acquired in one single event.

In Supplementary Fig. 18 we showed an example of the *mol_time* function clustering part output (*mclust* based) for sample PD26410d. For each clonally duplicated chromosome all the clonal SNVs are annotated according to their position (x-axis) and their corrected c-VAF (y-axis). All extracted clusters were highlighted with different colors. Gray dots represent the SNVs for which the cluster assignation was not possible due to insufficient certainty. On chromosome 11 we have three different clusters: SNVs on one allele (dark red), SNVs on two alleles (dark green), and SNVs on three (blue). The existence of three clusters, rather than of two, suggests that the first and second chromosome 11 gains were acquired in two independent MGs.

To avoid noise between the different clusters we applied a threshold below which data points that belong to an alternative cluster are excluded from the timing calculation. This second part is based on the assumption that all patients' chromosomes have a constant mutation rate (r). Using the *mol_time* function, we estimated the relative time of each chromosomal gain occurrence (molecular time; T) using the model and formulas summarized in Supplementary Fig. 19a and below:

Trisomy (2:1)

$$CN2 = r \times T$$
$$CN1 = r + 2 \times T \times (1 - T)$$
$$T = CN2/(CN2 + (CN1 - CN2))/3$$

Here CN2 and CN1 refers to the number of mutations detected on two and one alleles respectively. T and r refer to the molecular time and constant mutation rate respectively.

Changing conditions and formula, this approach may be extended also to other chromosomal duplications:

CN-LOH (2:0)

$$T = CN2/((CN2 + CN1)/2)$$

Gains with two extra copies (3:1).

$$T1st = CN3/(CN3 + CN2 + (CN1 - 2 \times CN2 - CN3)/4)$$
$$T2nd = (CN3 + CN2)/(CN3 + CN2 + (CN1 - 2 \times CN2 - CN3)/4))$$

Gains three extra copies (4:1)

$$T1st = (CN4 \times 5)/(CN1 + CN2 \times 2 + CN3 \times 3 + CN4 \times 4)$$
$$T2nd = ((CN4 + CN3) \times 5)/(CN1 + CN2 \times 2 + CN3 \times 3 + CN4 \times 4)$$
$$T3rd = ((CN4 + CN3 + CN2) \times 5)/(CN1 + CN2 \times 2 + CN3 \times 3 + CN4 \times 4)$$

Here CN3 and CN4 refers to the number of mutations detected on three and four alleles, respectively. T1st, T2nd, T3rd refer to the molecular time of the first, second, and third multi gain events.

Thanks to this approach we were able to estimate the molecular time of each copy-number duplication. In general, early gains will have a low ratio between duplicated (CN2) not-duplicated (CN1) mutations. Conversely late gains will have a high duplicated mutation burden (CN2) with a lower not-duplicated mutations (CN1) (Supplementary Fig. 19b).

Only clonal CNAs segments with a length > 1 Mb and a total number of clonal SNVs > 50 were considered. The confidence interval of each molecular time value was estimated using a bootstrapping function. To define if different gains occurred in one single time window or in different independent events we used multiple hierarchical clustering approach for each single bootstrap solution (*hclust* R function; www.r-project.org) and we integrated the most likely results with the Battenberg CNA changes over the time. To avoid any bias related to any potential subclonal mutation rate acceleration and heterogeneity, we included only clonal shared SNVs (early clonal) extracted by the Dirichlet process[5,38,53].

The recurrent MM CNAs chronological acquisition order was estimated combining the Battenberg CCF and molecular time data into a Bradley–Terry model, including just the earliest sample of each patient[23].

**Rearrangement timing.** Considering the number of reads supporting each rearrangement breakpoint and adjusting this value for both copy-number and CCF, we were able to estimate the adjusted VAF of each rearrangement (r-VAF). During any copy-number gain, the r-VAF of clonal SVs will change similarly to that of SNVs. Specifically, if the r-VAF is ~66% of all tumor reads, the SV will be classified as "pregain" being present on two different duplicated alleles. Conversely, if the r-VAF supports the involvement of just one allele (~33%), it may have occurred either on the not-duplicated allele or on one of the two duplicated alleles. We differentiated these two situations considering the status of any copy number generated by each SV and/or by the presence of in phase SNV/SNPs within any SV reads. The first approach is based on the fact that, if occurring after a gain, any deletion on the not-duplicated allele will generate an CN-LOH (2:0); and an involvement of one duplicated allele will generate a normal diploid segment (1:1). Conversely an involvement of the duplicated allele before the gain will generate a deletion (1:0) (Supplementary Fig. 20a, b).

The second approach is based on the presence of one or more clonal SNVs or SNPs phased within the reads supporting the rearrangement (Supplementary Fig. 20c, d). This specific event may generate three different situations:

(1) SNV/SNP c-VAF is ~66% and involves also all rearrangements reads (r-VAF ~66%). In this scenario, we can assume that the rearrangement and the substitution involved the duplicated allele before the gain.
(2) SNV/SNP c-VAF is ~66% and involves also all rearrangements reads (r-VAF ~33%). In this scenario, the rearrangement occurred on one of the two duplicated alleles after the gain.
(3) SNV/SNP c-VAF is ~33% and involved all rearrangements reads (r-VAF ~33%). In this case the rearrangement and the substitution occurred on the minor nonduplicated allele.

Merging these data together with the "molecular time", the CNA segment CCF and the structure of SV associated with CNAs were able to reconstruct the chronological order of many MM events.

**Digital PCR.** Droplet digital PCR assays were designed spanning the breakpoint of interest such that only mutant molecules would be amplified (Supplementary Table 1). PCR reactions were prepared in triplicate with 10 μl of 2× evagreen ddPCR supermix (Bio-Rad), 1 μl of assay mastermix (consisting of 16.2 μl H$_2$O, 14.4 μl of 25 μM forward primer, and 14.4 μl of 25 μM reverse primer) and 0.5 μl of whole genome amplified DNA in a total volume of 20 μl. PCR reactions were partitioned using a QX200 droplet generator according to the manufacturer's instructions. Samples were run on a PCR machine using the following parameters: 95 °C for 10 min, followed by 29 cycles of 94 °C for 30 s and 55 °C for 60 s, finally samples were held at 98 °C for 10 min and then kept at 4 °C until the next step. Plates were read on a Bio-Rad QX100 droplet reader and droplets positive for the assay target were quantified using QuantaSoft software (Bio-Rad).

## Data availability

Multiple myeloma sequence files are available at the European Genome-phenome archive under the accession codes EGAD00001003309 and EGAD00001001898. Data from the CoMpass study is available from dbGAP under the accession code phs000748.v1.p1.

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

## Acknowledgements

F.M. is supported by AIL (Associazione Italiana Contro le Leucemie-Linfomi e Mieloma ONLUS), by SIES (Società Italiana di Ematologia Sperimentale), and by the Memorial Sloan Kettering Cancer Center NCI Core Grant (P30 CA 008748). N.B. is funded by AIRC (Associazione Italiana per la Ricerca sul Cancro) through a MFAG (no. 17658) and by the European Research Council under the European Union's Horizon 2020 research and innovation program (grant agreement no. 817997). This work was supported by Department of Veterans Affairs Merit Review Award I01BX001584-01 (N.C.M.), NIH grants P01-155258 (N.C.M., H.A.L., M.F., P.J.C. and K.C.A.) and 5P50CA100707-13 (N.C.M., H.A.L. and K.C.A.).

## Author contributions

F.M., N.B. and P.J.C. designed the study, collected and analyzed the data, and wrote the paper; H.A.L., N.C.M. designed the study and collected the data; K.J.D., D.L., N.A., I.M., T.J.M., A.F., D.G., S.G., M.G., M.R., F.A., B.R.M., J.M.C.T. and D.C.W. analyzed the data; S.M., R.S., M.K.S., M.F., Y.T.T., M.FL., P.M., P.C. and K.C.A. collected the data.

## Additional information

**Competing interests:** The authors declare no competing interests.

