## [Peer Review File · Nature Communications]

Reviewers' comments:

Reviewer #1 (Remarks to the Author):

Maura et al. have made a significant effort to address the Reviewers' comments. This revised manuscript is significantly improved as it largely now focuses on the strengths of the study, however we wish to raise a few more points, which we hope will further increase the impact of this publication.

Major points:

- 1) We continue to believe that the most important contribution of this manuscript is molecular timing and the insight into the serial acquisition of gains (hyperdiploidy/1q). For quality control purposes, we would like to see the molecular timing algorithm applied on late samples only from patients with sequential samples available. The results can then be compared to what we know to be true, having already compared late to early samples. Does the algorithm successfully place new MM alterations in molecular time?
- 2) TET2, DNMT3A etc. are reported in this manuscript as significantly mutated in MM. These genes are frequently mutated at the germline level in patients with MM (Clonal Hematopoiesis of Indeterminate Potential). We would like the authors to confirm that these are indeed somatic and do not represent germline contamination. These are most likely germline contaminations.
- 3) What exactly is the contribution of the clustering analysis ("Dependencies of driver events")? It still takes up a big portion of Results, although it is not particularly memorable. The manuscript's Discussion is quite strong and persuasive, exactly because it focuses on the strengths of this work, which -the authors seem to agree- do not include the clustering analysis.
- 4) Have the authors looked at different trajectories, depending on treatment type?

Minor points:

- 1) The authors mention that TBC1D29 exhibits two hotspots, however, the lollipop plot does not support that claim, with only one mutation falling outside the only clear hotspot.
- 2) How did the authors decide on the SVs and CNAs to include in their clustering analysis (lines 161-163)?
- 3) Supplementary Figure 3 is low resolution and cannot be read properly.

Name of reviewer: Irene Ghobrial and Romanos Sklavenitis Pistofidis

Reviewer #2 (Remarks to the Author):

In their rebuttal, Maura et al. state that they have revised their manuscript "Genomic landscape and chronological reconstruction of driver events in multiple myeloma" to address reviewer concerns and improve how novel insights are presented.

In my opinion, the authors have unfortunately not been very responsive to reviewers' suggestions, as evidenced by the fact that almost all figure panels are unchanged.

Figure 1 is the same as in the original version. The authors acknowledge that 1B is difficult to understand but make essentially no changes. They acknowledge that they have not quantified the robustness of the clustering in 1c and that the survival analysis shows no difference between subgroups (with one exception). Therefore, as discussed previously, the significance of these data is highly uncertain, but the clustering and its discussion in the paper are largely unchanged. I am not sure to what degree including the code used for the cluster analysis in the submission is helpful here – the reader should be given a measure of robustness without having to repeat the

analysis on their own.

Figure 2 is largely the same with the exception of one panel. One of the concerns was that many of the figure panels show anecdotal examples and that it is not clear how these examples generalize to the patient cohort. Fig. 2c and d continue to consist of examples of chromothripsis and templated insertions in individual patents – I am not sure what we learn here.

Figure 3 is largely unchanged with some more individual examples added.

In Figure 4, the authors acknowledge that the robustness of their ordering (the central panel of the figure, 4b) is unclear, but the figure remains unchanged.

Figure 5 is unchanged and contains more individual examples.

Figure 6 is unchanged and wholly consists of individual examples without any summary plots that depict general patterns and their significance.

I understand that the review is now happening at a different journal and I am taking this into account. However, the central point of the original critique remains unaddressed: amidst the many individual examples, I do not see how the authors have clearly proven, with lucid figures that show informative statistics on the entire cohort, the general insights about multiple myeloma evolution that they claim to have obtained.

For example, the authors say in their rebuttal: “We described for the first time a rather gradual and slow acquisition of hyperdiploidy in multiple myeloma. This represents, in our opinion, a major insight, given that hyperdiploidy has previously been assumed to represent the consequence of a single catastrophic mitosis”. Which is the figure that proves that this is generally true? Figure 3c-e again shows individual examples which appear redundant with 4a. Figure 4a shows timing across the patient cohort, and in this figure it seems that the majority of gains happen in the “1st time window”, with relatively sporadic gains later. How does this figure really support the author’s claims?

The text above summarizes my evaluation of the revision. However, I am happy to acknowledge that perhaps some readers in the multiple myeloma field will find the individual examples presented in the figures compelling and illuminating. As a collection of interesting case studies, the paper could be suitable for publication in Nature Communications.

Reviewers' comments:

Reviewer #1 (Remarks to the Author):

Maura et al. have made a significant effort to address the Reviewers' comments. This revised manuscript is significantly improved as it largely now focuses on the strengths of the study, however we wish to raise a few more points, which we hope will further increase the impact of this publication.

Major points:

1) We continue to believe that the most important contribution of this manuscript is molecular timing and the insight into the serial acquisition of gains (hyperdiploidy/1q). For quality control purposes, we would like to see the molecular timing algorithm applied on late samples only from patients with sequential samples available. The results can then be compared to what we know to be true, having already compared late to early samples. Does the algorithm successfully place new MM alterations in molecular time?

We thank the reviewer for this suggestion. We agree that this analysis would significantly strengthen the confidence on the analysis. Considering all copy number gains acquired in serial samples in our cohort we observed different situations:

- Chromosomal segments with duplication of both the minor and the major allele. This condition doesn't allow a proper molecular time analysis because It is impossible to establish when -and on which allele- the mutation was duplicated;
- Subclonal gains: in this case the molecular time cannot be estimated because the algorithm works on clonal events;
- Clonal gains either acquired or lost after the treatment

For this last group we found 2 instances amenable to the analysis suggested. An example of this is the sample PD26412, where several gains were lost at relapse. For this case, the following selected gains were clonal in the first sample and therefore included in our standard molecular time analysis: a CN of 5 in chromosome 1q, a CN of 4 in chromosome 18, trisomies on chromosomes 2, 3, 7. At relapse, Chr 1q showed a CN of 4, Chr18 a CN of 3, and trisomies 2, 3 and 7 were subclonal. Our molecular time analysis on the first sample,

where these gains were seemingly clonal, showed a late and independent time window confirming the analysis is reliable.

This and another such example are now shown in the new Supplementary Figure 10-11 of the revised paper, and we also stressed this analysis in the revised manuscript at page 10-11, line 466-492 and in a new supplementary figure (Supplementary Figure 11)

2) TET2, DNMT3A etc. are reported in this manuscript as significantly mutated in MM. These genes are frequently mutated at the germline level in patients with MM (Clonal Hematopoiesis of Indeterminate Potential). We would like the authors to confirm that these are indeed somatic and do not represent germline contamination. These are most likely germline contaminations.

We thank the reviewer for this potentially intriguing observation. We tried to answer this question in two ways. First, we analysed the CCF of the 26 potential drivers from Walker et al, and we observed how a significant fraction of these mutations was characterized by a high cancer cell fraction, incompatible with contamination from myeloid cells bearing CHIP.

This is now included in the new supplementary figure 1 of the revised paper. Second, we checked metadata for the CoMMpass dataset, and observed how the source for germline material was peripheral blood in all cases. Given mutations were filtered with matched germline DNA in all patients, we would argue that CHIP mutations (also present in germline material) would be filtered out and not reported. We therefore conclude that the mutations observed are unlikely to be CHIP mutations arising from germline contamination, or from a CHIP HSC clone giving also rise to MM.

3) What exactly is the contribution of the clustering analysis ("Dependencies of driver events")? It still takes up a big portion of Results, although it is not particularly memorable. The manuscript's Discussion is quite strong and persuasive, exactly because it focuses on the strengths of this work, which -the authors seem to agree- do not include the clustering analysis.

The clustering analysis on WES data is relevant for at least a couple of reasons. First, the bayesian network allows the analysis of higher-level interactions between genomic events, and highlights how gene mutations are quite promiscuous and structural events take up most nodes of the network and of the subsequent clustering. Second, the clustering from 1C is used again in Figure 6 to group together the phylogenetic trees reconstructed from WGS: this shows that very few groups are homogeneous in their genomic makeup and modes of evolution. Therefore, we use the clustering analysis to justify the use of WGS in MM to capture more of the vast genomic complexity in MM, and to stress that exome studies can only explain little of the observed heterogeneity in MM.

In the revised paper, we have changed the text to better highlight the strengths and weaknesses of this analysis, and to help the reader to better understand its place in the flow of data. (page 14, line 590-605)

4) Have the authors looked at different trajectories, depending on treatment type?

While this would be highly relevant, unfortunately, we could not observe any specific evolutionary trajectory associated with a distinct treatment. This was mostly due to the limited sample size and to the heterogeneity in terms of treatment and clinical course (**Supplementary Table 1**).

Minor points:

1) The authors mention TBC1D29 that exhibits two hotspots, however, the lollipop plot does not support that claim, with only one mutation falling outside the only clear hotspot.

In the lollipop plot in Supplementary Figure 2 we did not show genomic positions on the X-axis for matters of space. However, the 4 positions where mutations accumulate in the last exon are 120 bp apart (~40 aminoacids), with the two recurrently mutated positions being 15 aminoacids apart. Whether this should be considered one or two hotspots is probably a bit of a subjective matter, but to avoid any confusion **we have removed this sentence in the revised text**.

2) How did the authors decide on the SVs and CNAs to include in their clustering analysis (lines 161-163)?

We manually selected the most recurrent events that are associated with distinct clinical and biological profiles based on literature review. **This sentence has now been included in the manuscript methods page 5 line 101-105; page 7 lines 359-363.**

3) Supplementary Figure 3 is low resolution and cannot be read properly.

A new Supplementary Figure 3 with high resolution has been provided in the revised manuscript.

Name of reviewer: Irene Ghobrial and Romanos Sklavenitis Pistofidis

Reviewer #2 (Remarks to the Author):

In their rebuttal, Maura et al. state that they have revised their manuscript “Genomic landscape and chronological reconstruction of driver events in multiple myeloma” to address reviewer concerns and improve how novel insights are presented.

In my opinion, the authors have unfortunately not been very responsive to reviewers’ suggestions, as evidenced by the fact that almost all figure panels are unchanged.

Figure 1 is the same as in the original version. The authors acknowledge that 1B is difficult to understand but make essentially no changes. They acknowledge that they have not quantified the robustness of the clustering in 1c and that the survival analysis shows no difference between subgroups (with one exception). Therefore, as discussed previously, the significance of these data is highly uncertain, but the clustering and its discussion in the paper are largely unchanged. I am not sure to what degree including the code used for the

cluster analysis in the submission is helpful here – the reader should be given a measure of robustness without having to repeat the analysis on their own.

Figure 2 is largely the same with the exception of one panel. One of the concerns was that many of the figure panels show anecdotal examples and that it is not clear how these examples generalize to the patient cohort. Fig. 2c and d continue to consist of examples of chromothripsis and templated insertions in individual patents – I am not sure what we learn here.

Figure 3 is largely unchanged with some more individual examples added.

In Figure 4, the authors acknowledge that the robustness of their ordering (the central panel of the figure, 4b) is unclear, but the figure remains unchanged.

Figure 5 is unchanged and contains more individual examples.

Figure 6 is unchanged and wholly consists of individual examples without any summary plots that depict general patterns and their significance.

I understand that the review is now happening at a different journal and I am taking this into account. However, the central point of the original critique remains unaddressed: amidst the many individual examples, I do not see how the authors have clearly proven, with lucid figures that show informative statistics on the entire cohort, the general insights about multiple myeloma evolution that they claim to have obtained.

For example, the authors say in their rebuttal: “We described for the first time a rather gradual and slow acquisition of hyperdiploidy in multiple myeloma. This represents, in our opinion, a major insight, given that hyperdiploidy has previously been assumed to represent the consequence of a single catastrophic mitosis”. Which is the figure that proves that this is generally true? Figure 3c-e again shows individual examples which appear redundant with 4a. Figure 4a shows timing across the patient cohort, and in this figure it seems that the majority of gains happen in the “1st time window”, with relatively sporadic gains later. How does this figure really support the author’s claims?

The text above summarizes my evaluation of the revision. However, I am happy to acknowledge that perhaps some readers in the multiple myeloma field will find the individual examples presented in the figures compelling and illuminating. As a collection of interesting case studies, the paper could be suitable for publication in Nature Communications.

We appreciate the negative tone of the reviewer, and we are sorry that the revised paper appeared rather unchanged to him/her. We felt his/her original review was very helpful, and we took care in trying to address the criticism received. The revised version initially submitted to nature communications consisted of an entirely new Figure 4, new supplemental figures, new panels, and text was edited in many paragraphs. For example:

- Figure 1b was particularly difficult to understand, and while this is inherent to the complex statistics used, in the revised version of the paper we i) modified the logic

gates to allow an easier interpretation ii) provided descriptive individual examples of some network subsections in supplementary to take the reader through the data shown.

- A completely new figure was created to take the reader through the changes over time of CNAs and complex structural events in the supplementary section.

However, we feel that continuing the list of changes would not help. Likely, we fell short of our objective. Rather than rebutting point-to-point the reviewer's claims, here we tried to further change the text and the appearance of some of the most problematic figures like figure 4 to provide some more details that may better support our claims and give a less anecdotal feel to the data shown, acknowledging the fair comments from the reviewer.

Ultimately, we stand by our data, and we believe our paper will represent a significant contribution to the field, particularly in the following aspects:

1. We have defined a genomic classification system for myeloma, based on current knowledge of all driver point mutations, SVs and copy number changes. This was based on innovative driver event discovery and clustering analysis, and shows strengths and weaknesses of exome-based clustering in suggesting new dimensions in the sample space compared to what currently known.
2. We have described a new mechanism of structural variation in myeloma, which contributes important driver events;
3. We have defined the timing of key copy number gains during evolution of hyperdiploidy, showing evidence for sequential acquisition of events;
4. We have provided the most detailed phylogenetic trees in multi-sampled myeloma from whole genome sequencing to date, showing interesting differences in timing of driver mutation acquisition across genomic subgroups.

REVIEWERS' COMMENTS:

Reviewer #1 (Remarks to the Author):

We believe the authors have addressed the Reviewers' comments and have no further suggestions. We look forward to seeing their work in print.

Reviewer #3 (Remarks to the Author):

I have been asked to comment specifically on the extent to which the authors have addressed the concerns raised by reviewer 2.

Overall I think the authors have done a reasonable job of addressing the criticisms raised by reviewer 2, and I do think on balance the manuscript should be published. However I would urge the authors to clarify certain issues raised by reviewer 2.

First, reviewer 2 raised the issue of whether the clustering (Figure 1c) is (i) reliable and (ii) clinically informative. In response the authors did make the clustering code available, but I am in agreement with reviewer 2 that a code dump is not necessarily the most helpful thing for the reader. Quoting reviewer 2 "the reader should be given a measure of robustness without having to repeat the analysis on their own". I agree with this. I think the authors should seriously consider including such a figure enabling the reader to get a sense of how robust the clustering is. On the issue of clinical relevance the authors state in their initial rebuttal that "The survival analysis shows no significant differences in survival between the groups". I believe it is important to be very upfront about this and would recommend the authors make a clear statement using words to that effect in the discussion. Because of this I would also argue that the claim made in the abstract that different evolutionary trajectories are "clinically relevant" be removed.

Second, regarding reviewer 2's other major criticism ("I do not see how the authors have clearly proven, with lucid figures that show informative statistics on the entire cohort, the general insights about multiple myeloma evolution that they claim to have obtained."), on balance I disagree with this statement. I do think the authors have shown that certain SV and CNA are early drivers of MM and this does seem like a novel, important insight. Though, given the authors do acknowledge a lack of robustness in the temporal ordering of events perhaps it should be stated more clearly in the

main text which events we can be confident occur early in MM evolution and which ones we are more uncertain about. I sympathise with Reviewer 2 that statements made in the rebuttal letter (“We described for the first time a rather gradual and slow acquisition of hyperdiploidy in multiple myeloma”) are perhaps a little bit of an over-claim since I agree with reviewer 2 that “majority of gains happen in the “1st time window”, with relatively sporadic gains later”. This could be ameliorated by the authors highlighting a little more clearly the limitations and uncertainties regarding the interpretations of the data.

Comments to the reviewers

Below a point-by-point response to the reviewer comments – the reviewer comments are in black, our response in blue and the actions we have taken in red.

Reviewer #1 (Remarks to the Author):

We believe the authors have addressed the Reviewers' comments and have no further suggestions. We look forward to seeing their work in print.

Reviewer #3 (Remarks to the Author):

I have been asked to comment specifically on the extent to which the authors have addressed the concerns raised by reviewer 2.

Overall I think the authors have done a reasonable job of addressing the criticisms raised by reviewer 2, and I do think on balance the manuscript should be published. However I would urge the authors to clarify certain issues raised by reviewer 2.

First, reviewer 2 raised the issue of whether the clustering (Figure 1c) is (i) reliable and (ii) clinically informative. In response the authors did make the clustering code available, but I am in agreement with reviewer 2 that a code dump is not necessarily the most helpful thing for the reader. Quoting reviewer 2 “the reader should measure of robustness without having to repeat the analysis on their own”. I agree with this. I think the authors should seriously consider including such a figure enabling the reader to get a sense of how robust the clustering is. On the issue of clinical relevance the authors state in their initial rebuttal that “The survival analysis shows no significant differences in survival between the groups”. I believe it is important to be very upfront about this and would recommend the authors make a clear statement using words to that effect in the discussion. Because of this I would also argue that the claim made in the abstract that different evolutionary trajectories are “clinically relevant” be removed.

To give the reader a quick idea of the robustness of clustering, in the revised version of the paper we have included a new supplementary figure 14 that summarizes the stability of the posterior probability of cluster assignment over 50,000 iterations.

Also, to tone down the claim on clinical relevance which is indeed not supported by data in the paper, we have removed that part of the sentence from the abstract

Second, regarding reviewer 2's other major criticism (“I do not see how the authors have clearly proven, with lucid figures that show informative statistics on the entire cohort, the general insights about multiple myeloma evolution that they claim to have obtained.”), on balance I disagree with this statement. I do think the authors have shown that certain SV and CNA are early drivers of MM and this does seem like a novel, important insight. Though, given the authors do acknowledge a lack of robustness in the temporal ordering of events perhaps it should be stated more clearly in the main text which events we can be confident occur early in MM evolution and which ones we are more uncertain about. I sympathise with Reviewer 2 that statements made in the rebuttal letter (“We described for the first time a rather gradual and slow acquisition of hyperdiploidy in multiple myeloma”) are perhaps a little bit of an over-claim since I agree

with reviewer 2 that “majority of gains happen in the “1st time window”, with relatively sporadic gains later”. This could be ameliorated by the authors highlighting a little more clearly the limitations and uncertainties regarding the interpretations of the data.

We agree that our analysis of the time windows of development of genetic lesions in MM life history has limitations: some stem from the relatively small sample size, others from an inherent limit of confidence of the analysis itself, which is clearly accounted for by confidence intervals in the data presented. Therefore, we have been extremely conservative in calling the different time windows. Nevertheless, according to our data, 13/18 HDMM patients showed CNAs acquired in different time windows. We consider this to be a remarkable fraction, and as the reviewer correctly states, the fact that trisomies and multiple copy gains are acquired over time even in late stages of myeloma development is one of the main conceptual advances of the paper.

In the revised version of the paper we have better highlighted the said limitations, but we also believe that some of our findings have immediate impact. Notably, the fact that trisomies in chr(1q) are among the earliest events, yet further copy-number gains of the same chromosome arm are late events fits well with the recently described “double-hit” myeloma category and has therefore clinical relevance. This is also now better stated in the text.